Identification and pathogenicity of Alternaria species associated with leaf blotch disease and premature defoliation in French apple orchards

Fontaine Kévin 1
Fourrier-Jeandel Céline 1
Armitage Andrew D. 2
Boutigny Anne-Laure 3
Crépet Manuela 4
Caffier Valérie 5
Gnide Dossi Carine 1
Shiller Jason 5
Le Cam Bruno 5
Giraud Michel 6
Ioos Renaud 1
Aguayo Jaime jaime.aguayo@anses.fr 1
1 ANSES, Laboratoire de la Santé des Végétaux, Unité de Mycologie, USC INRAE 1480 , Malzéville , France
2 Natural Resources Institute, University of Greenwich , Chatham Maritime , Kent , United Kingdom
3 ANSES, Laboratoire de la Santé des Végétaux, Unité Bactériologie, Virologie et OGM , Angers , France
4 FREDON Rhône-Alpes , Saint-Priest , France
5 Université d’Angers, Institut Agro, INRAE, IRHS, SFR QUASAV , Angers , France
6 Centre opérationnel de Lanxade, CTIFL , Prigonrieux , France
Adhikari Tika
Electronic publication date: 2021 Dec 1
Publication date: 2021
Volume: 9
Electronic Location ID: e12496
Received 2021 Jul 13; Accepted 2021 Oct 25
Copyright: ©2021 Fontaine et al.
Copyright year: 2021
Copyright holder: Fontaine et al.
License: This is an open access article distributed under the terms of the Creative Commons Attribution License, which permits unrestricted use, distribution, reproduction and adaptation in any medium and for any purpose provided that it is properly attributed. For attribution, the original author(s), title, publication source (PeerJ) and either DOI or URL of the article must be cited.
License URL: https://creativecommons.org/licenses/by/4.0/

Keywords: Alternaria leaf blotch, Alternaria fruit spot, Alternaria arborescens species complex, Alternaria alternata, Alternaria section Alternaria, Small-spored Alternaria, Plant pathology, Pathogenic fungi, Molecular biology, Pathogenicity of plant pathogens

Funding: French Ministry of Agriculture and the French Agency for Biodiversity ECOPHYTO/AFB AAP CASDAR French National Research Agency Laboratory of Excellence-ARBRE ANR-11-LABX-0002-01 This work was funded by the French Ministry of Agriculture and the French Agency for Biodiversity, ECOPHYTO/AFB, AAP CASDAR (call for projects 2017) recherche technologique 1716 CREATIVE The mycology research unit of the ANSES Plant Health Laboratory (LSV) is supported by a grant managed by the French National Research Agency (ANR) as part of the French government’s “Investing for the Future” (PIA) programme (ANR-11-LABX-0002-01, Laboratory of Excellence-ARBRE). The funders had no role in study design, data collection and analysis, decision to publish, or preparation of the manuscript.

==============================
Leaf blotch caused by Alternaria spp. is a common disease in apple-producing regions. The disease is usually associated with one phylogenetic species and one species complex, Alternaria alternata and the Alternaria arborescens species complex (A. arborescens SC), respectively. Both taxa may include the Alternaria apple pathotype, a quarantine or regulated pathogen in several countries. The apple pathotype is characterized by the production of a host-selective toxin (HST) which is involved in pathogenicity towards the apple. A cluster of genes located on conditionally dispensable chromosomes (CDCs) is involved in the production of this HST (namely AMT in the case of the apple pathotype). Since 2016, leaf blotch and premature tree defoliation attributed to Alternaria spp. have been observed in apple-producing regions of central and south-eastern France. Our study aimed to identify the Alternaria species involved in apple tree defoliation and assess the presence of the apple pathotype in French orchards. From 2016 to 2018, 166 isolates were collected and identified by multi-locus sequence typing (MLST). This analysis revealed that all these French isolates belonged to either the A. arborescens SC or A. alternata. Specific PCR detection targeting three genes located on the CDC did not indicate the presence of the apple pathotype in France. Pathogenicity was assessed under laboratory conditions on detached leaves of Golden Delicious and Gala apple cultivars for a representative subset of 28 Alternaria isolates. All the tested isolates were pathogenic on detached leaves of cultivars Golden Delicious and Gala, but no differences were observed between the pathogenicity levels of A. arborescens SC and A. alternata. However, the results of our pathogenicity test suggest that cultivar Golden Delicious is more susceptible than Gala to Alternaria leaf blotch. Implications in the detection of the Alternaria apple pathotype and the taxonomic assignment of Alternaria isolates involved in Alternaria leaf blotch are discussed.

Introduction

Alternaria spp. are ubiquitous fungi comprising approximately 300 different species (Simmons, 2007; Woudenberg et al., 2013). The genus has different lifestyles and can be isolated from a large number of substrates (Thomma, 2003; Woudenberg et al., 2013). Alternaria spp. are major pathogenic fungi in agriculture and the food industry, leading to serious diseases in many economically important crops (Meena et al., 2017; Thomma, 2003).

Different taxa of Alternaria have been associated with Alternaria leaf blotch (ALB) and Alternaria fruit spot (AFS) diseases (Armitage et al., 2015; Gur, Reuveni & Cohen, 2017; Harteveld, Akinsanmi & Drenth, 2013). ALB is characterized by the development of round or irregular brown spots on leaves, bordered by dark brown to purple margins (Rotondo et al., 2012). These symptoms generally start in late spring or early summer, developing to yellowing leaves that can lead to early tree defoliation and a premature fruit drop associated with a reduction in tree vigour and fruit quality over the following years (Harteveld, Akinsanmi & Drenth, 2013; Rotondo et al., 2012). ALB may cause up to 80% of defoliation in some susceptible apple cultivars (Filajdić & Sutton, 1991) and consequently may drastically decrease fruit yields (Harteveld, Akinsanmi & Drenth, 2013; Horlock, 2006). Less frequent, AFS is characterized by necrotic spots on the skin of the fruit surrounded by a red halo centered on the lenticels (Harteveld, Akinsanmi & Drenth, 2013; Horlock, 2006; Rotondo et al., 2012) and in some cases can result in calyx cracking and fruit rot (Gur, Reuveni & Cohen, 2017). AFS may consequently downgrade the fruit’s value, resulting in a significant financial burden to apple growers (Gur, Reuveni & Cohen, 2017; Harteveld, Akinsanmi & Drenth, 2014).

Both ALB and AFS have been reported in nearly all apple-producing regions of the world (Dickens & Cook, 1995; Filajdić & Sutton, 1991; Gur, Reuveni & Cohen, 2017; Harteveld, Akinsanmi & Drenth, 2013; Kim, Cho & Kim, 1986; Ozgonen & Karaca, 2006; Rotondo et al., 2012; Wenneker et al., 2018). Taxa causing ALB and AFS are part of the Alternaria section Alternaria that comprises the small-spored Alternaria species. As for the whole genus Alternaria, identification of isolates within the section Alternaria is challenging due to morphological plasticity and genetic similarity (Armitage et al., 2015; Lawrence et al., 2013; Woudenberg et al., 2013). However recent advances, especially in multi-gene phylogeny and comparative genomics, have allowed the different Alternaria sections to be redefined and delineated, with accurate molecular differentiation and identification of isolates (Armitage et al., 2020; Woudenberg et al., 2015a). Woudenberg et al. (2015a), for example, have shown that the Alternaria section Alternaria consists of 11 phylogenetic species and one species complex. The taxonomic implications of this study are major because 35 morphospecies, which could not be distinguished through multi-gene phylogeny, were synonymized under Alternaria alternata (including the important plant pathogens A. alternata, A. tenuissima and A. citri).

ALB and AFS have been commonly associated with the phylogenetic species A. alternata and the Alternaria arborescens species complex (A. arborescens SC) (Gur, Reuveni & Cohen, 2017; Harteveld, Akinsanmi & Drenth, 2013; Rotondo et al., 2012; Toome-Heller et al., 2018; Wenneker et al., 2018), with both taxa also known as saprophytic and generalist opportunistic pathogens affecting a variety of important crops (Armitage et al., 2015; Thomma, 2003). Both A. alternata and A. arborescens SC may include the apple pathotype, which has been recently shown to be polyphyletic (Armitage et al., 2020). The apple pathotype, formerly known as Alternaria mali, causes significant problems in apple orchards in south-eastern Asia (Li et al., 2019), was responsible for ALB in the south-eastern USA in the early nineties (Filajdić & Sutton, 1991) and has been associated with severe AFS in Israel (Gur, Reuveni & Cohen, 2017). It is also listed either as a quarantine or a regulated pathogen in several countries throughout the world (https://gd.eppo.int/).

In Alternaria, pathotypes are characterized by the production of polyketide host-selective toxins (HSTs), which are linked to pathogenicity affecting specific hosts (Tsuge et al., 2013). To date, at least seven pathotypes have been described, each producing a unique HST essential to pathogenicity in apples (AMT), Japanese pears (AKT), strawberries (AFT), tangerines (ACT), tomatoes (AAL) rough lemons (ACR) and tobacco (AT) (Tsuge et al., 2013; Wang et al., 2019). The production of these HSTs involves a cluster of genes located on conditionally (or accessory) dispensable chromosomes (CDCs), so named because they are not essential for saprophytic growth and reproduction of pathogens (Hatta et al., 2002; Wang et al., 2019).

In the case of the apple pathotype, at least 17 genes could be involved in the synthesis of AMT apple toxin (Harimoto et al., 2007) but so far only four, i.e., AMT1, AMT2, AMT3 and AMT4, have been demonstrated to be involved in this process. To date, molecular detection of this pathotype is only possible by PCR targeting one of the genes involved in the production of the AMT apple toxin (Armitage et al., 2020; Harimoto et al., 2007; Johnson et al., 2000) or by identifying these genes in the genome of Alternaria isolates using bioinformatics (Armitage et al., 2020). However, molecular taxonomic assignment to either A. alternata or the A. arborescens SC in routine diagnostics requires the construction of multi-gene phylogenies (Armitage et al., 2015; Harteveld, Akinsanmi & Drenth, 2013; Rotondo et al., 2012; Woudenberg et al., 2015a). Indeed, multi-locus sequence typing (MLST) with relevant phylogenetic markers for the Alternaria genus has been used to identify A. alternata and the A. arborescens SC. It has also enabled researchers to understand their association with ALB and AFS in several countries (Armitage et al., 2015; Gur, Reuveni & Cohen, 2017; Harteveld, Akinsanmi & Drenth, 2013; Rotondo et al., 2012; Toome-Heller et al., 2018; Wenneker et al., 2018).

ALB has been observed for years in French orchards without causing serious damage. However, since 2016, significant defoliation in trees infected by Alternaria has been reported in regions of central and south-eastern France. The reported presence of the apple pathotype in northern Italy (Rotondo et al., 2012) has raised serious concerns for both the French plant health authorities and apple growers.

The first objective of this study was to assess whether the upsurge in ALB symptoms observed in French orchards was due to the emergence of the apple pathotype. To identify the pathogens responsible for these unusual cases of defoliation, we conducted MLST analyses to determine the phylogenetic position of the French Alternaria isolates. We also assessed the presence of the apple pathotype by PCR tests targeting two genes involved in the production of the AMT apple toxin, and a gene found in the apple pathotype CDC that has homologs in the pear and strawberry pathotypes (Armitage et al., 2020). Pathogenicity tests were then carried out and Koch’s postulates were assessed on the Gala and Golden Delicious apple cultivars using a representative panel of isolates. Finally, both the phylogenetic position and pathogenicity of the French Alternaria isolates were compared with Alternaria isolates from different countries and/or isolated from crops other than apple.

Materials & Methods

Isolate collection

French Alternaria spp. isolates (166) were obtained from symptomatic field samples of leaves (156) and fruit (10) collected from ten different apple cultivars or cultivar groups (Table 1, Supplemental Information 1). Apple orchards were located in four major apple-producing regions in central and south-eastern France: Auvergne-Rhône-Alpes (108 isolates), Provence-Alpes-Côtes d’Azur (49 isolates), Occitanie (8 isolates) and Nouvelle-Aquitaine (1 isolate). The samples were collected over 3 years (seasons 2016–17, 2017–18 and 2018–19, Table 1, Supplemental Information 1). Leaves and fruit surfaces were first disinfected with 70% ethanol and necrotic spots were excised using a sterile scalpel blade then plated onto Petri dishes containing malt extract agar (Sigma-Aldrich) medium supplemented with chloramphenicol (0.2 g/L). The cultures were incubated for four to seven days at 22 °C with a 12 h alternating dark and light cycling period. A plug of each actively growing culture was then transferred to a new malt extract agar Petri dish and incubated under the same conditions as described above. The isolate collection was supplemented with 43 Alternaria isolates either associated with ALB or AFS from Australia (16), Israel (eight), Italy (14) and New Zealand (five). Furthermore, the Food and Environmental Research Agency (FERA) contributed 12 isolates obtained from fruit importations showing AFS and intercepted in the UK (unknown origin). Nine additional ALB or AFS isolates were obtained from the Westerdijk Institute’s collection (https://www.wi.knaw.nl). Other Alternaria isolates associated with post-harvest apple rot problems in Argentina (four) and South Africa (eight) were also included. Additionally, the collection was completed with Alternaria isolates from other hosts belonging to different botanical families (19 isolates). Five isolates in the collection were identified as apple pathotype isolates. All the isolates were single-spored before analysis. Details of the isolate collection studied in this work are presented in Supplemental Information 1.

Table 1 Distribution of Alternaria isolates obtained from French orchards from years 2016–2018.

The table shows the apple cultivar, the number of samples, the taxa (Alternaria arborescens species complex (SC) or Alternaria alternata) and the co-occurrence of the isolates in the same orchard. The samples were identified by sequencing EndoPg and Alta-1.

Cultivar	No. of samples (including fruit samples)	A. arborescens SC	A. alternata	Co-occurrence/sample (including fruit samples)	
Braeburn	2	4	4	1	
Galaa	13 (2)	16	8	3	
Golden Delicious	10	39	10	5	
Canadab	9	14	17	3	
Dalinette	4	5	12	4	
Crimsonc	4	2	8	1	
Belchard	1	1	1	1	
Pink Ladyd	7 (3)	9	10	4 (2)	
Garance	1	0	2	0	
GoldRush	1	1	3	1	
Total	52 (5)	91	75	23	
Notes.

a cultivars Galastar and Royal Gala included.

b cultivar Reinette grise du Canada (Canada).

c cultivar Crimson Crisp included.

d cultivar Rosy Glow included.

Genomic DNA extraction and loci sequencing

DNA extractions were performed with approximately 0.5 g of Alternaria mycelium, scraped from a fresh culture on malt extract, using the NucleoSpin Plant II kit (Macherey Nagel). Mycelium was ground by placing two sterilized steel beads (three mm in diameter) in an Eppendorf tube containing mycelium with 400 µL of lysis buffer and 10 µL of RNAse (both provided with the DNA extraction kit). Samples were subsequently ground twice for 60 s at 30 Hz in a MM 400 mixer mill (Retch). DNA was extracted according to the manufacturer’s instructions. The concentration of the DNA extracts (100 µL final volume) was estimated with a NanoDrop TM 2000 Spectrophotometer (ThermoFisher). All the extracted DNA was stored at −30 °C until use. The endopolygalacturonase (EndoPG) and the Alternaria major allergen (Alta-1) genes, two loci commonly used in Alternaria identification and phylogenetics (Table 2, Armitage et al., 2015; Harteveld, Akinsanmi & Drenth, 2013; Lawrence et al., 2013), were sequenced for all the isolates. In addition, the anonymous region OPA 10-2 (Table 2, Rotondo et al., 2012; Woudenberg et al., 2015a; Woudenberg et al., 2015b) was sequenced for a subset of 100 isolates and used to assess putative differences in taxonomic identification by comparison with EndoPG and Alta-1. For PCR amplification of the three loci, the reaction mixtures contained 1X PCR reaction buffer (HGS Diamond Taq, Eurogentec), 2.5 mM MgCl2 (4.0 mM for EndoPG), 4 × 0.25 mM dNTPs, 0.2 µM of forward and reverse primers (Table 2), 1 U of HGS diamond Taq (Eurogentec), 2 µL of DNA extract and molecular grade water to complete up to 25 µL. PCR conditions consisted of an initial denaturation step at 95 °C for 10 min, followed by 40 cycles at 94 °C for 45 s, annealing temperatures of 57 °C for Alta-1, 56 °C for EndoPG and 62 °C for OPA 10-2 for 30 s (Table 2), 72 °C for 1 min and a final extension step at 72 °C for 7 min. The GENEWIZ sequencing platform (Leipzig, Germany) was used for bidirectional Sanger sequencing of the amplicons. Consensus sequences were obtained after manual correction using the Geneious R11 programme.

Table 2 Characteristics of primer pairs used in this study for multi-locus sequence typing (MSLT) identification of isolates and specific PCR.

Locus/function	Primer	Primer sequence (5′–3′)	Reference	Annealing temperature (°C)	Amplicon length (pb)	
Alta-1/Alternaria major allergen 1	Alt-for	ATGCAGTTCACCACCATCGC	Hong et al. (2005)	57	472	
	Alt-rev	ACGAGGGTGAYGTAGGCGTC				
EndoPG/Endopolygalacturonase	PG3	TACCATGGTTCTTTCCGA	Andrew, Peever & Pryor (2009)	56	464	
	PG2b	GAGAATTCRCARTCRTCYTGRTT				
OPA 10-2/Anonymous
noncoding region	OPA10-2L	TCGCAGTAAGACACATTCTACG	Andrew, Peever & Pryor (2009)	62	634	
	OPA10-2R	GATTCGCAGCAGGGAAACTA				
AMT-1/Non-ribosomal peptide synthethase	LinF1	TATCGCCTGGCCACCTACGC	Johnson et al. (2000)	65	496	
	LinR	TGGCCACGACAACCCACATA				
AMT-2/Aldo-keto reductase	AMT2-f2	GTTGCAGAATCGCAAACTCA	Roberts, Bischoff & Reymond (2012)	57	653	
	AMT2-r2	GGCTCTTGGTCTCAAATCCA				
AMT-14/Unknown function	AMT14-EMR-F	TTTCTGCAACGGCGKCGCTT	Armitage et al. (2015)			
	AMT14-EMR-R	TGAGGAGTYAGACCRGRCGC		66	436	

Phylogenetic analysis

The EndoPG, Alta-1 and OPA 10-2 sequence datasets generated in our study were supplemented with data from previous studies. The sequence datasets that enabled taxonomic identification of isolates in these previous studies (Armitage et al., 2015; Gur, Reuveni & Cohen, 2017; Harteveld, Akinsanmi & Drenth, 2013; Rotondo et al., 2012; Woudenberg et al., 2015a) were used as a reference in our phylogenetic analysis. As we performed molecular identification using an MLST approach, we decided to use the taxonomy of the Alternaria section Alternaria proposed by Woudenberg et al. (2015a), which consists of 11 phylogenetic species and one species complex. In other words, we used the Alternaria alternata phylogenetic species without including any results of morphospecies (e.g., A. tenuissima was taxonomically assigned to the phylogenetic species A. alternata), an approach used in other studies that described isolates morphologically (Armitage et al., 2015; Rotondo et al., 2012). DNA sequences were first analyzed with SeaView version 4 (Gouy, Guindon & Gascuel, 2010). These analyses included sequence alignments using MUSCLE (Edgar, 2004) and elimination of poorly aligned positions with Gblocks (Talavera & Castresana, 2007). MrBayes version 3.2 (Ronquist et al., 2012b) was used for multi-locus phylogeny analysis on concatenated sequences for EndoPG and Alta-1 (two-locus MSLT phylogenetic tree) and EndoPG, Alta-1 and OPA 10-2 (three-locus MLST phylogenetic tree) separately. Runs were performed under the Bayesian MCMC model jumping approach, which provides a convenient alternative to model selection before analysis (command lset applyto= (all) nst=mixed). In model jumping, the Markov Chain Monte Carlo (MCMC) sampler explores different models and weights the results according to the posterior probability of each model (Ronquist, Huelsenbeck & Teslenko, 2012a). Four MCMC chains were run using the default heating with tree sampling performed every 5,000 generations. Runs were performed for at least 20 million generations, and stopped when the standard deviation of split frequencies was below 0.01 (Ronquist, Huelsenbeck & Teslenko, 2012a). Homologous sequences of Alta-1 and Endo-PG for A. brassicicola (isolate Abra43) were used as an outgroup in all the generated trees. The consensus tree was obtained by using the command sumt. The resulting phylogenetic trees were visualised and annotated with the interactive tree of life (iTOL) online tool (Letunic & Bork, 2016). The taxonomic identification of 100 isolates using the concatenated trees EndoPg/Alta-1 and EndoPg/Alta-1/OPA 10-2 was compared with the function tanglegram implemented in DENDROSCOPE 3.2.10 (Huson & Scornavacca, 2012). A subset of single-locus sequence data for the corresponding loci was submitted to Genbank (accession nos. MN975269 –MN975340, Supplemental Information 1).

PCR detection of the Alternaria apple pathotype

We searched for the Alternaria apple pathotype among all the French isolates by PCR targeting two genes involved in AMT apple toxin biosynthesis—namely AMT1 and AMT2—using primers developed by Johnson et al. (2000) and Harimoto et al. (2007) respectively (Table 2). Additionally, 44 out of these French isolates were also tested by PCR targeting AMT14, a gene found in the apple pathotype toxin gene cluster and for which homologous genes also exist in pear and strawberry pathotypes (Armitage et al., 2020). Other non-French isolates were tested by PCR targeting either AMT1/AMT2 or AMT1/AMT2/AMT14 genes (27 and 47 isolates respectively, Supplemental Information 1). PCRs were performed in 25-µL reaction mixtures containing 1X PCR reaction buffer (HGS Diamond Taq, Eurogentec), 2.5 mM MgCl2, 4 × 0.25 mM dNTPs, 0.2 µM of forward and reverse primers (Table 2), 1 U of HGS diamond Taq (Eurogentec), 2 µL of DNA extract and molecular grade water to complete up to 25 µL. PCR conditions comprised an initial 10 min denaturation step at 95 °C followed by 40 cycles of a denaturation step at 94 °C for 30 s, an annealing step at 65 °C for AMT1, 57 °C for AMT2, 66 °C for AMT14 (Table 2) and an extension step at 72 °C for 60 s. These cycles were followed by a final extension at 72 °C for 7 min. All PCRs were performed in duplicate. Controls were included in all reactions. Positive controls included either gDNA (Alternaria apple pathotype isolate LSVM 75) for AMT14 testing or a plasmid solution of AMT1 and AMT2 genes inserted in a vector using the pCR4-TOPO cloning kit (Invitrogen) following the manufacturer’s instructions. Negative controls consisted of sterile distilled water (SDW).

Figure 1 Bayesian phylogenetic tree of Alta-1 and EndoPg markers.

The tree was constructed with sequences of 352 Alternaria isolates (261 sequences were generated in this study). The color legend refers to the Bayesian posterior probabilities of the tree nodes. Alternaria alternata isolates are shown in blue. The Alternaria arborescens SC isolates are shown in red. Isolates from other taxonomic groups of the Alternaria section Alternaria are represented in orange (Alternaria alstroemeriae), purple (Alternaria gaisen), brown (Alternaria longipes) and grey (Alternaria gossypina). Isolates used in pathogenicity tests are highlighted in a yellow background.

Pathogenicity assays and Koch’s postulates

Twenty-eight isolates were used for pathogenicity assays: ten A. alternata, 17 A. arborescens SC and one A. brassicicola (Supplemental Information 1). Their selection took into account isolation from different apple cultivars (10) and origin (5 countries) (Supplemental Information 1). It also covered different clades or subclades of the MLST phylogenetic tree constructed with the EndoPG and Alta-1 regions (Fig. 1). No apple pathotype isolate could be tested, because no strain in our collection could produce a sufficient amount of conidia in culture, despite several attempts. The A. brassicicola isolate Abra43 was included as a non-pathogenic control. Assays were performed on detached apple leaves from Golden Delicious clone X972 and Gala clone X4712, the most industrially-relevant cultivars in France, representing 45% of the total apple production in the country (AGRESTE, 2021). For simplicity, both clones will be referred hereafter to as Golden Delicious and Gala. Spore suspensions were obtained from isolates grown at 22 °C for 21 days on malt extract agar medium under specific light condition cycles (Carvalho et al., 2008): an incubation period of 7 days under a 12 h alternating dark then light cycling period followed by 2 days under an 8 h-UV/16 h dark conditions (UV light-induced by a black fluorescent near UV lamp (Philips/15W, T8-BLB) and a final cycle of 12 days in full darkness. For each isolate, the inoculum was obtained by flooding the culture with 2 mL of SDW before dislodging spores by scraping the plate with an L-shaped spreader. After a filtration step with sterile gauze, the spore suspensions were counted and adjusted to a concentration of 1 × 105 conidia/mL with a haemocytometer. Leaves from the third or fourth node were detached from fresh branches of apple saplings grown in a glasshouse. Eight leaves cleaned with 70% ethanol were placed in plastic boxes containing two white absorbent paper towels humidified with SDW and conserved at ambient temperature overnight before leaf inoculations. An experimental replicate consisted of one strain inoculated on five different leaves (placed in five different plastic boxes) per cultivar. Unwounded abaxial leaf surfaces were inoculated at six points with 10 µL of conidial suspension. Each plastic box contained a negative control that consisted of one leaf inoculated with SDW. Inoculated leaves were incubated at 20 °C for 10 days under an alternating 12 h dark then light cycling period. Each isolate was tested twice in independent experiments. The results from each experiment were analyzed separately at three data collection times: 4, 7 and 10 days post-inoculation (dpi). Two types of analysis were performed. For data from 4 dpi, a zero-inflated Poisson general linear mixed model (GLMM) was used to assess the number of lesions per leaf. The model included the following explanatory variables: the taxon of the tested isolate (A. alternata or A. arborescens SC), the apple cultivar (Gala or Golden Delicious), and an experiment repeat variable (each isolate was tested twice in independent experiments). An isolate effect was taken into account as a random variable. For data from 7 and 10 dpi, zero-inflated beta GLMMs were performed. On both 7 and 10 dpi the response variable was the proportion of the diseased leaf area on detached leaves that corresponds to the lesion size. The diseased leaf area proportion (necrosis) was assessed by visual inspection and coded between 0 and 1. The model included the same explanatory and random effect variables used for the 4 dpi data model: taxon, cultivar, repeat and isolate (random effect). All GLMM analyses (on 4, 7 and 10 dpi) took into account only isolates of A. alternata and A. arborescens SC, as only one A. brassicicola isolate was used, which is not enough to be included in the isolate random effect. All the models were run in the R environment (version 4.0.3) using the glmmTMB package (Brooks et al., 2017). Excess zeros were in all cases tested with the function testZeroInflation of the DHARMa R package (Hartig, 2017), which compares the distribution of expected zeros in the data with the observed zeros. Model residual diagnostics of all models were performed with the DHARMa package. Analysis of variance type II (ANOVA type II) was performed to assess the effect of each explanatory variable. Koch’s postulates were assessed by the re-isolation on malt agar extract medium of 23 randomly chosen tested isolates (Supplemental Information 1) and the re-sequencing of Alta-1 and EndoPG loci.

Results

Molecular identification of strains

Concatenation of Alta-1 and Endo-PG sequences resulted in a 900-bp alignment. This alignment was used for phylogenetic analyses. Depending on the isolate, the number of bases/residues that differed between isolates of A. alternata and isolates of other taxa of the Alternaria section Alternaria included in the analysis (A. arborescens SC, A. gaisen, A. longipes, A. gossypina and A. alstroermeriae) ranged from 6 to 37. The two-marker phylogenetic tree distinguished two major clades: A. alternata and A. arborescens SC (Fig. 1). The A. alternata phylogenetic clade encompassed four subclades, while the A. arborescens SC encompassed two. The analysis could also distinguish these two clades from other taxa in the Alternaria section Alternaria: A. gaisen, A. longipes, A. gossypina and A. alstroemeriae (Fig. 1). The three concatenated genes (Alta-1/Endo-PG/OPA10-2) resulted in a 1,534-bp alignment which included 18–65 differences/residues between A. alternata and other taxa of the Alternaria section Alternaria included in the analysis. As for the two-marker concatenated tree, the phylogenetic analysis using three markers distinguished two major clades, A. alternata and A. arborescens SC (Supplemental Information 2). It could also distinguish these clades from other taxa in the Alternaria section Alternaria (A. gaisen, A. longipes, A. gossypina and A. alstroemeriae). The phylogenetic tree pattern was similar to that determined with two loci (i.e., Alta-1 and EndoPG). The A. alternata clade was divided into four subclades, whereas the A. arborescens SC was separated into two subclades. Adding a third locus to the analysis (OPA 10-2) did not improve the resolution within A. alternata and A. arborescens SC (Supplemental Information 2). The two phylogenetic analyses did, however, refine identification of two isolates: two Australian strains were consequently assigned to A. alternata whereas they had previously been assigned to A. longipes (BRIP46356 and BRIP46455) by Harteveld, Akinsanmi & Drenth (2013) (Fig. 1, Supplemental Information 1).

Identification of Alternaria isolates causing ALB in France

Isolates from France were identified as either A. arborescens SC (91 isolates, 55%) or A. alternata (75 isolates, 45%) based on the taxonomic identification with the Alta-1 and EndoPG markers. No changes in the taxonomic identification were observed for the subset of isolates with sequences of Alta-1, EndoPG and OPA 10-2 markers (Supplemental Information 1). The distribution of isolates differed according to the cultivar (Table 1). It was observed that A. arborescens SC isolates were more frequent on Gala and Golden Delicious cultivars, whereas A. alternata isolates were more frequent on cultivars Reinette grise du Canada, Dalinette and Crimson. On Braeburn and Pink lady cultivars, there was a similar number of isolates of A. arborescens SC and A. alternata. On Garance and GoldRush (Coop38cov) cultivars, too few isolates were recovered to make any comparison (Table 1).

Screening for the Alternaria apple pathotype

None of the 166 French isolates were identified as the apple pathotype by PCR tests targeting the AMT apple toxin (AMT1, AMT2) or cross-pathotype (AMT14) loci (Supplemental Information 1). All five apple pathotype reference isolates behaved as expected and yielded positive results for AMT1 and AMT2 PCR tests and also for the tests targeting AMT14, which is common to apple, pear and strawberry pathotypes (Supplemental Information 1). However, four isolates formerly identified as apple pathotype in earlier studies in Italy (Rotondo et al., 2012) and Israel (Gur, Reuveni & Cohen, 2017) gave negative PCR results for the three loci in our conditions, thus overturning their identification (Supplemental Information 1).

Pathogenicity assays and Koch’s postulates

At 4 dpi, 27 isolates—including the negative control A. brassicicola—were able to induce at least one necrotic spot on detached leaves of apple cultivars Golden Delicious and Gala. The only exception was an A. arborescens SC isolate (16_489b3a) that did not induce necrotic spots on detached Gala cultivar leaves. The zero-inflated Poisson GLMM used to assess the number of lesions per leaf on 4 dpi, indicated a significant effect of the experiment repetition (Type II Wald; χ2 = 17.75; df = 2; p = 0.00014) but not of the taxa (Type II Wald; χ2 = 2.44; df = 1; p = 0.118) or the apple cultivar (Type II Wald; χ2 = 0.06; df = 1; p = 0.810) (Fig. 2). All 28 tested isolates were able to induce leaf blotch after 7 dpi on Golden Delicious and Gala. On 7 dpi, the zero-inflated beta GLMM used to assess the proportion of the diseased leaf area indicated a significant effect of the experiment repetition (Type II Wald; χ2 = 84.013; df = 2; p < 0.001) and the apple cultivar (Type II Wald; χ2 = 17.296; df = 1; p = 3.199e−5), but not of the taxa (Type II Wald; χ2 = 0.001; df = 1; p = 0.97) (Fig. 3A). On 10 dpi the zero-inflated beta GLMM indicated a significant effect of the experiment repetition (Type II Wald; χ2 = 91.82; df = 2; p < 2.2e−16) and the apple cultivar (Type II Wald; χ2 = 11.80; df = 1; p = 0.0006), but not of the taxa (Type II Wald; χ2 = 0.06; df = 1; p = 0.801) (Fig. 3B). On 7 and 10 dpi, leaves of the Golden Delicious cultivar were more susceptible than those of Gala, as measured by the proportion of the diseased leaf area (Fig. 3). Raw measurements at 4, 7 and 10 dpi are presented in the Supplemental Information 4 section. Finally, the identity of 23 of these strains was confirmed by re-isolating and sequencing (Alta-1 and EndoPG loci), fulfilling Koch’s postulates (Supplemental Information 1). No disease symptoms were observed on leaves inoculated with water. Examples of the results from the pathogenicity tests are shown in the Supplemental Information 5 section.

Figure 2 Mean number of leaf lesions per cultivar (Gala in red and Golden Delicious in blue) 4 days post-inoculation (4 dpi).

Results are reported for isolates of Alternaria alternata, Alternaria arborescens SC and Alternaria brassicicola, which were identified by multi-locus sequence typing (MLST). Pathogenicity tests were performed on unwounded abaxial leaf surfaces with six separate point inoculations of 10 µl of Alternaria conidial suspensions (concentration of 1 × 105 conidia/µL). Statistical tests were only performed on Alternaria alternata and Alternaria arborescens SC. No differences were observed between isolates of Alternaria alternata and Alternaria arborescens SC or the apple cultivar (Golden Delicious and Gala). A significant effect of the experiment repetition (Type II Wald; χ2 = 17.75; p < 0.001) was observed.

Figure 3 Mean proportion of the diseased leaf area per cultivar 7 and 10 days post-inoculation (7, 10 dpi) reported for Alternaria alternata, Alternaria arborescens SC and Alternaria brassicicola.

(A) Mean proportion of the diseased leaf area per cultivar (Gala in red and Golden Delicious in blue) 7 days post-inoculation (7 dpi) reported for isolates of Alternaria alternata, Alternaria arborescens SC and Alternaria brassicicola, identified by multi-locus sequence typing (MLST). Statistical tests were only performed on Alternaria alternata and Alternaria arborescens SC. The results showed that on 7 dpi, leaves of the Golden Delicious cultivar were more susceptible than leaves of the Gala cultivar (Type II Wald; χ2 = 17.296; p < 0.001). No differences were observed between isolates of Alternaria alternata and Alternaria arborescens SC. A significant effect of the experiment repetition (Type II Wald; χ2 = 84.013; p < 0.001) was also observed. (B) Mean proportion of the diseased leaf area per cultivar (Gala in red and Golden Delicious in blue) 10 days after inoculation (10 dpi) reported for isolates of Alternaria alternata, Alternaria arborescens and Alternaria brassicicola, identified by multi-locus sequence typing (MLST). Statistical tests were only performed on Alternaria alternata and Alternaria arborescens SC. The results showed that on 10 dpi, leaves of the Golden Delicious cultivar were more susceptible than leaves of the Gala cultivar (Type II Wald; χ2 = 11.80; p < 0.001). No differences were observed between isolates of Alternaria alternata and Alternaria arborescens SC. A significant effect of the experiment repetition (Type II Wald; χ2 = 91.82; p < 0.001) was also observed.

Discussion

The Alternaria apple pathotype was not found in French orchards

We firstly checked whether we were witnessing the emergence of the apple pathotype in French orchards. We did this through PCR assays targeting three genes located in the conditionally dispensable chromosome (CDC)—AMT1 (Johnson et al., 2000), AMT2 (Harimoto et al., 2008) and AMT14 (Armitage et al., 2020)—which characterizes isolates of the apple pathotype. Our results showed that this pathogen was not present in France within the sampled regions and years. To date, molecular detection of the apple pathotype has only been possible by PCR tests that target genes present in the CDC. These targets are associated with secondary metabolite clusters involved in the production of the apple pathotype host-specific toxin AMT (Armitage et al., 2020; Harimoto et al., 2007; Johnson et al., 2000). Although all our apple pathotype reference strains gave positive results using the three markers, the Italian and Israeli strains previously identified as apple pathotype by PCR based on the amplification of a modified PCR test targeting AMT1 (Rotondo et al., 2012) and AMT3 (Gur, Reuveni & Cohen, 2017) gave negative results in our study with tests targeting loci AMT1, AMT2 and AMT14. In the case of the Israeli strains, the primers targeting AMT3 (Harimoto et al., 2007), showed unsatisfactory results in our preliminary tests as several unexpected bands appeared after gel electrophoresis of the PCR product (data not shown) and these primers were discarded for subsequent molecular tests. The results obtained with the Italian strains are more difficult to explain because the initial study of Rotondo et al. (2012) performed several confirmation tests (including sequencing of the products). One hypothesis that may explain the difficulty in amplifying these loci is the occurrence of partial or total chromosomal loss in isolates. This phenomenon has previously been reported in the apple pathotype by Johnson et al. (2001) and is due to chromosomal instability in culture. To avoid this problem, in the analysis of French isolates, our tests targeting AMT1, AMT2 and AMT14 were performed right after isolation, avoiding several subculturing cycles. However, in all the cases where the presence of AMT1, AMT2 and AMT14 was assessed (in the five reference isolates; Supplemental Information 1), all three gene-specific PCR assays gave positive results. Based on the results of our study, we suggest that the apple pathotype should be detected from pure cultures by using at least two or more of the existing molecular tests to target AMT1 and AMT2, which is a good option if the objective is to specifically detect the apple pathotype. By optimizing and validating current tools or developing new molecular tests, it might be possible to detect diseases in planta from symptomatic leaves, which could avoid isolate subculturing cycles while minimizing the risk of chromosomal loss.

Co-existence of Alternaria alternata and the Alternaria arborescens species complex in French orchards

The second objective of this study was to identify Alternaria species or groups associated with Alternaria leaf blotch (ALB) and Alternaria fruit spot (AFS) in French orchards. The phylogenetic trees generated after using MLST clearly showed that these diseases are caused by two phylogenetic clades: A. alternata and A. arborescens SC, regardless of the apple cultivar. Our results also showed that both taxa may co-exist in the same orchard. These results confirm that these two Alternaria taxa are the major cause of ALB and AFS in regions of the world where these diseases have been studied so far (Gur, Reuveni & Cohen, 2017; Harteveld, Akinsanmi & Drenth, 2013; Rotondo et al., 2012; Toome-Heller et al., 2018; Wenneker et al., 2018). In addition, our results suggest that sequencing two loci, i.e., Alta-1 and EndoPG, is enough to be able to distinguish Alternaria isolates involved in these diseases. Firstly, these two loci enable the two major phylogenetic clades—A. alternata and A. arborescens SC—to be distinguished. Secondly, the loci also clearly distinguish these two clades from other Alternaria taxa within the Alternaria section. Including the OPA 10-2 locus did not substantially improve the molecular identification of the strains.

Alternaria alternata and the Alternaria arborescens SC are responsible for defoliation in French apple orchards

We showed that the isolates collected from necrotic leaves were able to produce symptoms on detached apple leaves of cultivars Gala and Golden Delicious. The latter cultivar was more susceptible under our conditions, as shown by measurements of the diseased leaf area after 7 and 10 dpi, a quantitative trait generally used to measure pathogen aggressiveness. Gala and Golden Delicious were used for pathogenicity tests since they are the most important cultivars in France. Although these cultivars are considered as relatively “susceptible” to ALB and AFS (Filajdić & Sutton, 1991; Harteveld, Akinsanmi & Drenth, 2014; Rotondo et al., 2012), earlier studies have shown that there is little, or no cultivar specificity in Alternaria taxa causing ALB and AFS, at least for the most economically important apple cultivars used worldwide (Filajdić & Sutton, 1991; Harteveld, Akinsanmi & Drenth, 2014; Li et al., 2019). Management of the disease may involve resistant cultivars. However, further research involving more cultivars would be required to assess potential cultivar-specificity among Alternaria taxa causing ALB and AFS. The selection of disease-resistant cultivars should rely upon studies combining data collected from the field and from trials under controlled conditions (Li et al., 2019).

Our results also showed that the entire subset of Alternaria isolated from apple leaves or fruit fulfilled Koch’s postulates. The pathogenicity tests showed that there are no significant differences between isolates of A. alternata and A. arborescens SC as assessed by the number of lesions per leaf on 4 dpi or the proportion of the diseased leaf area on 7 and 10 dpi. Both results are in agreement with previous studies that suggest that pathogenicity may be isolate-dependent rather than species-dependent (Harteveld, Akinsanmi & Drenth, 2014; Rotondo et al., 2012). One of the limits of our study is that we could not assess the pathogenicity of any of the reference apple pathotype strains because too few spores could be obtained during cultivation. It is important to highlight, however, that previous studies comparing the pathogenicity of apple pathotype isolates with other Alternaria isolates in apples have shown discrepant results: while Armitage et al. (2020) showed that apple pathotype strains were significantly more pathogenic than other isolates that do not carry CDCs, Rotondo et al. (2012) did not observe any difference in levels of pathogenicity between apple pathotype isolates and other Alternaria isolated from apple leaves or fruit. Unexpectedly, we observed symptoms on apple leaves inoculated by A. brassicicola, which has never been reported as pathogenic on apples to our knowledge. These results suggest that Alternaria isolates from other Alternaria sections that do not carry CDCs involved in the production of HTS may also cause ALB symptoms under controlled conditions. This is probably associated with the production of nonspecific Alternaria toxins that can affect many plants regardless of whether they are or are not a host of the pathogen (Tsuge et al., 2013). However, as shown here and elsewhere, under natural conditions only small-spore Alternaria (Alternaria section Alternaria) have so far been described as apple pathogens causing ALB and AFS. Recent genomic resources, including the genome of A. brassicicola (Belmas et al., 2018) and isolates of Alternaria involved in AFS and ALB (Armitage et al., 2020) will allow comparative genomics analysis that may clarify these pathogenicity mechanisms.

Finally, this study identified the Alternaria taxa involved in ALB and AFS in France, but did not determine the cause of the increased severity in these diseases over recent years (e.g., introduction of the apple pathotype). However, alternative explanations may be suggested based on previous epidemiological studies. Firstly, it seems that the disease develops better in relatively hot (between > 20 °C and 30 °C) and rainy weather (Bhat, Peerzada & Anwar, 2015; Filajdić & Sutton, 1992; Harteveld et al., 2013; Kim, Cho & Kim, 1986). Potential changes in these two parameters, or other climatic factors, should be studied in greater depth in the French regions concerned by AFS. Another hypothesis is the introduction of more virulent strains. This could occur by the long-distance movement of spores carried by wind currents that may have transported Alternaria air inoculum into apple orchards from sources in other apple-producing regions (Fernández-Rodríguez et al., 2015; Woudenberg et al., 2015b). Finally, the emergence of fungicide resistance among strains should not be ruled out, considering that apple orchards are treated intensely with fungicides, mainly used to control apple scab caused by Venturia inaequalis, which also contributes to the control of ALB and AFS (Horlock, 2006).

Conclusions

Since 2016, Alternaria leaf blotch and premature defoliation attributed to Alternaria spp. have been observed in apple-producing regions in central and south-eastern France. The emergence of the Alternaria apple pathotype was suspected following its observation in northern Italy. The presence of the apple pathotype in French orchards was therefore assessed by a specific PCR targeting three genes located on conditionally dispensable chromosomes across a large collection of Alternaria isolates. Our results showed that the Alternaria apple pathotype was not present. Taxonomic identification of these isolates, assessed by multi-locus sequence typing and construction of phylogenetic trees, indicates that Alternaria leaf blotch in France is associated with isolates of A. alternaria and A. arborescens SC. Pathogenicity tests of a subsample of isolates demonstrated that they were all able to induce necrotic symptoms on detached apple leaves of the cultivars Gala and Golden Delicious. Our results also showed that there are no significant differences in levels of pathogenicity between isolates of A. alternata and A. arborescens SC. Our controlled pathogenicity tests do suggest, however, that cultivar Golden Delicious is more susceptible to Alternaria leaf blotch. In the future, genetic and epidemiological approaches are required to clarify why Alternaria leaf blotch events have increased in frequency and severity in some regions of France.

Supplemental Information

Supplemental Information 1 List and characteristics of isolates used in this study

Accessions in bold and red indicate the sequences that were submitted to Genbank. Accessions in blue indicate that the isolate has the same sequence that the one submitted (indicated in red) to Genbank. In the column pathogenicity, √ indicates the isolates for which Koch’s postulates were assessed by re-isolation and re-sequencing. In the column AMT1/AMT2/AMT14, 0 = no PCR amplification of the target gene, 1 = positive amplification of the target gene, - = not tested.

Click here for additional data file.

Supplemental Information 2 Bayesian phylogenetic tree of Alta-1, EndoPg and OPA 10-2 markers

The tree was constructed with sequences of 100 Alternaria isolates. The color legend refers to the Bayesian posterior probabilities of the tree nodes. Alternaria alternata isolate names are shown in blue. Alternaria arborescens SC isolates are shown in red. Isolates from other taxonomic groups of the Alternaria section Alternaria are represented in orange (Alternaria alstroemeriae), purple (Alternaria gaisen), brown (Alternaria longipes) and grey (Alternaria gossypina).

Click here for additional data file.

Supplemental Information 3 Comparison of cladograms obtained from Bayesian phylogenetic trees of Alta-1 and EndoPg markers and Alta-1, EndoPg and OPA 10-2 markers

The analysis included a subset of 100 isolates. Isolate clades are represented by different colors: blue for Alternaria alternata, red for Alternaria arborescens SC and grey for Alternaria gaisen, Alternaria alstroemeriae, Alternaria longipes and Alternaria gossypina. The figure only shows some of the isolate names.

Click here for additional data file.

Supplemental Information 4 Raw measurements data of the number of lesions per leaf at 4 days post-inoculation (dpi) and the proportion of the diseased leaf area at 7 and 10 dpi

Data are presented by isolate, taxon, repetition and the apple cultivar (Gala or Golden Delicious)

Click here for additional data file.

Supplemental Information 5 Examples of a single experimental replicate used in pathogenicity tests 4, 7 and 10 days post-inoculation (dpi)

Inoculation tests with detached leaves of the Gala apple cultivar. The codes next to each leaf are isolate codes. Isolates of Alternaria arborescens are shown in red and isolates of Alternaria alternata in purple.

(B) Inoculation tests with detached leaves of the Golden Delicious apple cultivar. The codes next to each leaf are isolate codes. Isolates of Alternaria arborescens are shown in red and isolates of Alternaria alternata in purple.

Pathogenicity tests were performed by inoculation of unwounded abaxial leaf surfaces with 10 µL of conidial suspensions (concentration of 1 × 105 conidia/mL). An experimental replicate consisted of one strain inoculated on five different leaves (placed in five different plastic boxes) per cultivar.

Click here for additional data file.

We would also like to thank the French apple producers who supplied leaf and fruit samples, and are grateful to Pr. Thomas Guillemette, Pr. Barry Pryor, Pr. Andrea Patriarca, Dr. Olufemi A. Akinsanmi, Dr. Moshe Reuveni, Dr. Lior Gur, Dr. Marina Collina and Ms. Jacqueline Hubert for sharing Alternaria strains. We are also grateful to the LSV ANSES team members who helped to perform some of the experiments, in particular Maurane Pagniez for the purification of single-spore strains and Eugenie Vuittenez for the pathogenicity assays.

Additional Information and Declarations

Competing Interests

Author Contributions

DNA Deposition

Data Availability

The authors declare there are no competing interests.

Kévin Fontaine, Céline Fourrier-Jeandel, Dossi C. Gnide, Renaud Ioos and Jaime Aguayo are employed by ANSES, Malzéville, France.

Anne-Laure Boutigny is employed by ANSES, Angers, France.

Manuela Crepet is employed by FREDON Rhône-Alpes, France.

Michel Giraud is employed by Centre opérationnel de Lanxade, CTIFL, France.

Kévin Fontaine, Céline Fourrier-Jeandel and Dossi Carine Gnide performed the experiments, analyzed the data, authored or reviewed drafts of the paper, and approved the final draft.

Andrew D. Armitage, Anne-Laure Boutigny, Manuela Crépet, Valérie Caffier and Jason Shiller performed the experiments, authored or reviewed drafts of the paper, and approved the final draft.

Bruno Le Cam, Michel Giraud and Renaud Ioos conceived and designed the experiments, authored or reviewed drafts of the paper, and approved the final draft.

Jaime Aguayo conceived and designed the experiments, performed the experiments, analyzed the data, prepared figures and/or tables, authored or reviewed drafts of the paper, and approved the final draft.

The following information was supplied regarding the deposition of DNA sequences:

A subset of single-locus sequence data for the corresponding loci are available at Genbank: MN975269 to MN975340.

The following information was supplied regarding data availability:

The raw measurements are available in the Supplementary Files.

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
