# Peer review of "Identification and pathogenicity of Alternaria species associated with leaf blotch disease and premature defoliation in French apple orchards"

_PeerJ, doi:10.7717/peerj.12496_

## Round 0.1 · original submission · Major Revisions

Dear authors,

Your manuscript which you submitted to PeerJ has been reviewed. The comments of the reviewer(s) are included at the bottom of this letter or attached for your reference and revision.

Based on the comments of the reviewers and my own assessment, your manuscript will require MAJOR revisions prior to further consideration for publication. I am also reserving the right to send your revised manuscript out to the original reviewer(s) or to the new reviewer(s) before making a final decision on suitability for publication.

In addition to two reviewers' comments, please check the attached pdf file for my MAJOR comments (yellow-highlighted areas).

Once again, thank you for submitting your manuscript to PeerJ and I look forward to receiving your revision.

Best regards,

Sincerely,

Tika Adhikari

·

Basic reporting

The article is well written and easy to read. While the introduction is slightly on the longer side, it does contain important information to build an understanding for the next sections. I would suggest that the section would be divided into smaller paragraphs as it currently appears to have only two paragraphs. Smaller sections would make it easier to follow the text and separate the different topics.
The manuscript is well structured and illustrated with a range of informative figures and tables. Adding the supplementary data provides further information which might also be useful to some readers.

Experimental design

The study included a large number of samples from a range of geographic areas, collected over three years. That results in a sufficiently representative collection to make conclusions about the pathogen populations and authors have provided sufficient detail about the methodology they have used for sequence data generation and analysis, as well as for pathogenicity testing.

Validity of the findings

The results and conclusions from the study are well supported and presented. Publishing these results will be useful to a range of professionals working in apple orchards. It is important to make these findings publicly available so we could better understand the role of different Alternaria species in leaf blotch and fruit spot disease.

Additional comments

There are a few minor aspects that would be good to revisit and clarify. Please see some small suggestions below:
- Line 142 - could you please clarify what do you mean by "malt medium"? Was this malt extract agar?
- Line 288 – the first part of the sentence reads a bit odd. Maybe “resulted in a 900 bp long alignment” or “Concatenated Alta-1 and Endo-PG were in total 900 bp long”?
- Line 303 – remove word “only”
- Line 352 – modify the sentence as follows “… examples of the results from the pathogenicity tests…”
- The introductory section in the beginning of the discussion (lines 356-360) do not add much value, in my opinion. I would suggest removing it and moving straight to discussing the results.
- I would also suggest to remove the conclusions section as it is very similar to the abstract and the well structured discussion section already allows to find the three main points about the findings of this study.

·

Basic reporting

The manuscript is very well written. Literature references are adequate. Very well structured manuscript.

Experimental design

The experiments are very well carried out. No further comments.

Validity of the findings

The data are robust and sound. Conclusions are clear. No further comments

Additional comments

Dear authors, a very good manuscript. I have made a number of suggestions in the attached pdf. These are mainly edits (e.g. cultivar instead of variety).

---

## Round 0.2 · Minor Revisions

Dear Dr. Ioos and Dr. Aguayo,

Thank you for revising and submitting your manuscript to PeerJ. I still noticed some errors and clarity in your manuscript. I would appreciate it if you could read the author guidelines of the PeerJ and reformat your whole manuscript accordingly. For example, some words such 'analyse to analyze' 'characterise to characterize', 'programme' to 'program'. Please see the attached file for few examples.

Importantly, extensive editing is highly recommended for the clarity of your manuscript.

Thank you for your understanding.

Best regards,

Tika Adhikari

---

## Round 0.3 · accepted · Accept

Dear Dr. Ioos and Dr. Aguayo,

I am writing to inform you that your manuscript - Identification and pathogenicity of Alternaria species associated with leaf blotch disease and premature defoliation in French apple orchards - has been Accepted for publication. Congratulations!

Thank you for your submission to PeerJ.

Best regards.


Sincerely,


Tika Adhikari